# ENHANCED CONVOLUTIONAL NEURAL KERNELS

## ABSTRACT

Recent research shows that for training with $\ell_2$ loss, convolutional neural networks (CNNs) whose width (number of channels in convolutional layers) goes to infinity correspond to regression with respect to the CNN Gaussian Process kernel (CNN-GP) (Novak et al., 2019) if only the last layer is trained, and correspond to regression with respect to the Convolutional Neural Tangent Kernel (CNTK) if all layers are trained. An exact algorithm to compute CNTK (Arora et al., 2019) yielded the finding that classification accuracy of CNTK on CIFAR-10 is within 6-7% of that of the corresponding CNN architecture (best figure being around 78%) which is interesting performance for a fixed kernel.

Here we show how to significantly enhance the performance of these kernels using two ideas. (1) Modifying the kernel using a new operation called *Local Average Pooling* (LAP) which preserves efficient computability of the kernel and inherits the spirit of standard data augmentation using pixel shifts. Earlier papers were unable to incorporate naive data augmentation because of the quadratic training cost of kernel regression. This idea is inspired by *Global Average Pooling* (GAP), which we show for CNN-GP and CNTK is equivalent to full translation data augmentation. (2) Representing the input image using a pre-processing technique proposed by Coates et al. (2011), which uses a single convolutional layer composed of random image patches.

On CIFAR-10, the resulting kernel, CNN-GP with LAP and horizontal flip data augmentation, achieves 89% accuracy, matching the performance of AlexNet (Krizhevsky et al., 2012) , and outperforms the best previous classifier that is not a trained neural network (Mairal, 2016). Similar improvements are obtained for Fashion-MNIST.

## 1 INTRODUCTION

Recent research shows that for training with $\ell_2$ loss, convolutional neural networks (CNNs) whose width (number of channels in convolutional layers) goes to infinity, correspond to regression with respect to the CNN Gaussian Process kernel (CNN-GP) if only the last layer is trained (Novak et al., 2019; Garriga-Alonso et al., 2019), and correspond to regression with respect to the Convolutional Neural Tangent Kernel (CNTK) if all layers are trained (Jacot et al., 2018; Allen-Zhu et al., 2018; Du et al., 2019b; Arora et al., 2019). Novak et al. (2019); Garriga-Alonso et al. (2019) also implemented CNN-GP and tested its empirical performance. An efficient exact algorithm was given (Arora et al., 2019) to compute CNTK for CNN architectures, as well as those that include a *Global Average Pooling* (GAP) layer (defined below). This is a fixed kernel that inherits some benefits of CNNs, including exploitation of locality via convolution, as well as multiple layers of processing. For CIFAR-10, incorporating GAP into the kernel improves classification accuracy by up to 10% compared to pure convolutional CNTK.

While this performance is encouraging for a fixed kernel, the best accuracy is still under 78%, which is disappointing even compared to AlexNet. One hope for improving the accuracy further is to somehow capture modern innovations such as batch normalization, data augmentation, residual layers, etc. in CNTK. The current paper shows how to incorporate simple data augmentation. Specifically, the idea of creating new training images from existing images using pixel translation and flips, while assuming that these operations should not change the label. Since deep learning uses stochastic gradient descent (SGD), it is trivial to do such data augmentation on the fly. However, it's

unclear how to efficiently incorporate data augmentation in kernel regression, since training time is quadratic in the number of training images. [1]

Thus somehow data augmentation has to be incorporated into the computation of the kernel itself. The main observation here is that the above-mentioned algorithm for computing CNTK involves a dynamic programming whose recursion depth is equal to the depth of the corresponding finite CNN. It is possible to impose symmetry constraints at any desired layer during this computation. In this viewpoint, it can be shown that prediction using CNTK/CNN-GP with GAP is equivalent to prediction using CNTK/CNN-GP without GAP but with *full translation data augmentation* with wrap-around at the boundary. The translation invariance property implicitly assumed in data augmentation is exactly equivalent to an imposed symmetry constraint in the computation of the CNTK which in turn is derived from the pooling layer in the CNN. See Section 4 for more details.

Thus GAP corresponds to full translation data augmentation scheme, but in practice such data augmentation creates unrealistic images (cf. Figure 2) and training on them can harm performance. However, the idea of incorporating symmetry in the dynamic programming leads to a variant we call *Local Average Pooling* (LAP). This implicitly is like data augmentation where image labels are assumed to be invariant to small translation, say by a few pixels. Interestingly, LAP corresponds to a average pooling layer for CNNs, named box filtering (Szeliski, 2010).

Experimentally, we find LAP significantly enhances the performance as discussed below.

- In extensive experiments on CIFAR-10 and Fashion-MNIST, we find LAP consistently improves performance of CNN-GP and CNTK. In particular, we find CNN-GP with LAP achieves $81\%$ on CIFAR-10 dataset, outperforming the best previous kernel predictor by $3\%$.
- When using the technique proposed by Coates et al. (2011), which uses randomly sampled patches from training data as filters to do pre-processing,[2] CNN-GP with LAP and horizontal flip data augmentation achieves $89\%$ accuracy on CIFAR-10, matching the performance of AlexNet (Krizhevsky et al., 2012) and is the strongest classifier that is not a trained neural network.[3]
- We also test performance of CNNs with an extra layer corresponding to LAP and observe that it improves the performance on certain architectures.

## 2 RELATED WORK

Data augmentation has long been known to improve the performance of neural network and kernel methods (Sietsma & Dow, 1991; Schölkopf et al., 1996). Theoretical study of data augmentation dates back to Chapelle et al. (2001). Recently, Dao et al. (2018) proposed a theoretical framework for understanding data augmentation and showed data augmentation with a kernel classifier can have feature averaging and variance regularization effects. More recently, Chen et al. (2019) quantitatively shows in certain settings, data augmentation provably improves the classifier performance. For more comprehensive discussion on data augmentation and its properties, we refer readers to Dao et al. (2018); Chen et al. (2019) and references therein.

CNN-GP and CNTK correspond to infinitely wide CNN with different training strategies (only training the top layer or training all layers jointly). The correspondence between infinite neural networks and kernel machines was first noted by Neal (1996). More recently, this was extended to deep and convolutional neural networks (Lee et al., 2018; Matthews et al., 2018; Novak et al., 2019; Garriga-Alonso et al., 2019). These kernels correspond to infinitely wide neural networks where only the last layer is trained. A recent line of work studied overparameterized neural networks where all layers are trained (Allen-Zhu et al., 2018; Du et al., 2019b; 2018; Li & Liang, 2018; Zou et al., 2018). Their proofs imply the gradient kernel is close to a fixed kernel which only depends the training data and neural network architecture. These kernels thus correspond to infinitely wide neural networks where are all layers are trained. Jacot et al. (2018) named this kernel, neural tangent kernel (NTK). Arora et al. (2019) formally proved infinitely wide neural net predictor trained by

---

[1]For CIFAR 10, the bottleneck of using CNN-GP and CNTK is not solving least square but constructing kernel. The time complexity is $O(p^2 n^2)$ where $p$ is the number of pixels in each image and $n$ is number of data point.

[2]See Section 6.2 for the precise procedure.

[3]https://benchmarks.ai/cifar-10

gradient descent is equivalent to NTK predictor. Recently, NTKs induced by various neural network architectures are derived and shown to achieve strong empirical performance (Arora et al., 2019; Yang, 2019; Du et al., 2019a).

Global Average Pooling (GAP) was first proposed in Lin et al. (2013) and is common in modern CNN design (Springenberg et al., 2014; He et al., 2016; Huang et al., 2017). However, current theoretical understanding on GAP is still rather limited. It has been conjectured in Lin et al. (2013) that GAP reduces the number of parameters in the last fully-connected layer and thus avoids overfitting, and that GAP is more robust to spatial translations of the input since it sums out the spatial information. In this work, we study GAP from the CNN-GP and CNTK perspective, and draw an interesting connection between GAP and data augmentation.

Here we are interested in methods that are not trained neural networks. If the features are predefined before seeing the data, Oyallon & Mallat (2015) proposed the scattering network which achieves 82% classification accuracy on CIFAR-10. If one uses unsupervised learning methods to extract features, the method proposed in Coates et al. (2011) is one of the best-performing approaches on CIFAR-10 preceding modern CNNs. To our knowledge, the best result via unsupervised learning method in this line is by Mairal (2016), who used the convolutional kernel network to achieve 86% accuracy on CIFAR-10. In this work we combine CNTK with LAP and the idea in Coates et al. (2011) to achieve the best performance for classifiers that are not trained neural networks.

## 3 PRELIMINARIES

### 3.1 NOTATION

We use bold-faced letters for vectors, matrices and tensors. For a vector $\boldsymbol{a}$, let $[\boldsymbol{a}]_i$ be its $i$-th entry; for a matrix $\boldsymbol{A}$, let $[\boldsymbol{A}]_{i,j}$ be its $(i,j)$-th entry; for an order 4 tensor $\boldsymbol{T}$, let $[\boldsymbol{T}]_{ij,i'j'}$ be its $(i,j,i',j')$-th entry. For a symmetric tensor, wet let $\text{tr}(\boldsymbol{T}) = \sum_{i,j} \boldsymbol{T}_{ij,ij}$. For an order $d$ tensor $\boldsymbol{T} \in \mathbb{R}^{C_1 \times C_2 \times \dots \times C_d}$ and an integer $\alpha \in [C_d]$, we use $\boldsymbol{T}_{(\alpha)} \in \mathbb{R}^{C_1 \times C_2 \times \dots \times C_{d-1}}$ to denote the order $d-1$ tensor formed by fixing the coordinate of the last dimension of $\boldsymbol{T}$ to be $\alpha$.

### 3.2 CNN, CNN-GP AND CNTK

In this section we give formal definitions of CNN, CNN-GP and CNTK that we study in this paper. Throughout the paper, we let $P$ be the width and $Q$ be the height of the image. We use $q \in \mathbb{Z}_+$ to denote the filter size. In practice, $q = 1, 3, 5$ or $7$.

**Padding Schemes.** In the definition of CNN, CNTK and CNN-GP, we may use different padding schemes. Let $\boldsymbol{x} \in \mathbb{R}^{P \times Q}$ be an matrix. For a given index pair $(i,j)$ with $i \leq 0$, $i \geq P + 1$, $j \leq 0$ or $j \geq Q + 1$, different padding schemes define different value for $[\boldsymbol{x}]_{i,j}$. For *circular padding*, we define $[\boldsymbol{x}]_{i,j}$ to be $[\boldsymbol{x}]_{i \bmod P, j \bmod Q}$. For *zero padding*, we simply define $[\boldsymbol{x}]_{i,j}$ to be 0. Note the difference between circular padding and zero padding occurs only on the boundary of images. We will prove our theoretical results for the circular padding scheme to avoid boundary effects.

**CNN.** Now we describe CNN with and without GAP. For any input image $\boldsymbol{x}$, after $L$ intermediate layers, we obtain $\boldsymbol{x}^{(L)} \in \mathbb{R}^{P \times Q \times C^{(L)}}$ where $C^{(L)}$ is the number of channels of the last layer. See Section A for the definition of $\boldsymbol{x}^{(L)}$. For the output, there are two choices: with and without GAP.

- Without GAP: the final output is defined as $f(\boldsymbol{\theta}, \boldsymbol{x}) = \sum_{\alpha=1}^{C^{(L)}} \left\langle \boldsymbol{W}_{(\alpha)}^{(L+1)}, \boldsymbol{x}_{(\alpha)}^{(L)} \right\rangle$ where $\boldsymbol{x}_{(\alpha)}^{(L)} \in \mathbb{R}^{P \times Q}$, and $\boldsymbol{W}_{(\alpha)}^{(L+1)} \in \mathbb{R}^{P \times Q}$ is the weight of the last fully-connected layer.

- With GAP: the final output is defined as $f(\boldsymbol{\theta}, \boldsymbol{x}) = \frac{1}{PQ} \sum_{\alpha=1}^{C^{(L)}} \boldsymbol{W}_{(\alpha)}^{(L+1)} \cdot \sum_{(i,j) \in [P] \times [Q]} \left[ \boldsymbol{x}_{(\alpha)}^{(L)} \right]_{i,j}$ where $\boldsymbol{W}_{(\alpha)}^{(L+1)} \in \mathbb{R}$ is the weight of the last fully-connected layer.

**CNN-GP and CNTK.** Now we describe CNN-GP and CNTK. Let $\boldsymbol{x}, \boldsymbol{x}'$ be two input images. We denote the $L$-th layer's CNN-GP kernel as $\boldsymbol{\Sigma}^{(L)}(\boldsymbol{x}, \boldsymbol{x}') \in \mathbb{R}^{[P] \times [Q] \times [P] \times [Q]}$ and the $L$-th layer's CNTK kernel as $\boldsymbol{\Theta}^{(L)}(\boldsymbol{x}, \boldsymbol{x}') \in \mathbb{R}^{[P] \times [Q] \times [P] \times [Q]}$. See Section A for the precise definitions of

$\mathbf{\Sigma}^{(L)}(\boldsymbol{x}, \boldsymbol{x}')$ and $\mathbf{\Theta}^{(L)}(\boldsymbol{x}, \boldsymbol{x}')$. For the output kernel value, again, there are two choices, without GAP (equivalent to using a fully-connected layer) or with GAP.

- Without GAP: the output of CNN-GP is $\mathbf{\Sigma}_{\mathsf{FC}}(\boldsymbol{x}, \boldsymbol{x}') = \operatorname{tr}\left(\mathbf{\Sigma}^{(L)}(\boldsymbol{x}, \boldsymbol{x}')\right)$ and the output of CNTK is $\mathbf{\Theta}_{\mathsf{FC}}(\boldsymbol{x}, \boldsymbol{x}') = \operatorname{tr}\left(\mathbf{\Theta}^{(L)}(\boldsymbol{x}, \boldsymbol{x}')\right)$.
- With GAP: the output of CNN-GP is $\mathbf{\Sigma}_{\mathsf{GAP}}(\boldsymbol{x}, \boldsymbol{x}') = \frac{1}{P^2 Q^2}\sum_{i,j,i',j'\in[P]\times[Q]\times[P]\times[Q]}\left[\mathbf{\Sigma}^{(L)}(\boldsymbol{x}, \boldsymbol{x}')\right]_{i,j,i',j'}$, and the output of CNTK is $\mathbf{\Theta}_{\mathsf{GAP}}(\boldsymbol{x}, \boldsymbol{x}') = \frac{1}{P^2 Q^2}\sum_{i,j,i',j'\in[P]\times[Q]\times[P]\times[Q]}\left[\mathbf{\Theta}^{(L)}(\boldsymbol{x}, \boldsymbol{x}')\right]_{i,j,i',j'}$.

**Kernel Prediction.** Lastly, we recall the formula for kernel regression. For simplicity, throughout the paper, we will assume all kernels are invertible. Given a kernel $\mathbf{K}(\boldsymbol{x}, \boldsymbol{x}')$ and a dataset $(\boldsymbol{X}, \boldsymbol{y})$ with data $\{(\boldsymbol{x}_i, y_i)\}_{i=1}^N$, define $\mathbf{K}_{\mathbf{X}} \in \mathbb{R}^{N\times N}$ where $[\mathbf{K}_{\mathbf{X}}]_{i,j} = \mathbf{K}(\boldsymbol{x}_i, \boldsymbol{x}_j)$. The prediction for unseen data $\boldsymbol{x}'$ is $\sum_{i=1}^N \alpha_i \mathbf{K}(\boldsymbol{x}', \boldsymbol{x}_i)$, where $\boldsymbol{\alpha} = \mathbf{K}_{\mathbf{X}}^{-1}\boldsymbol{y}$.

### 3.3 Data Augmentation Schemes

In this paper we consider two types of data augmentation schemes: translation and horizontal flip.

**Translation.** Given $(i,j)\in[P]\times[Q]$, we define the translation operator $\mathcal{T}_{ij} : \mathbb{R}^{P\times Q\times C} \to \mathbb{R}^{P\times Q\times C}$: for an image $\boldsymbol{x}\in\mathbb{R}^{P\times Q\times C}$, $[\mathcal{T}_{ij}(\boldsymbol{x})]_{i',j',c} = [\boldsymbol{x}]_{i'+i,j'+j,c}$ for $(i',j',c)\in[P]\times[Q]\times[C]$. Here the precise definition of $[\boldsymbol{x}]_{i'+i,j'+j,c}$ depends on the padding scheme. Given a dataset $D = \{(\boldsymbol{x}_i, y_i)\}_{i=1}^N$, the *full translation data augmentation scheme* creates a new dataset $D_{\mathcal{T}} = \{(\mathcal{T}_{ij}(\boldsymbol{x}_i), y_i)\}_{(i,j,n)\in[P]\times[Q]\times[N]}$ and training is performed on $D_{\mathcal{T}}$.

**Horizontal Flip.** Flip operator $\mathcal{F} : \mathbb{R}^{P\times Q\times C} \to \mathbb{R}^{P\times Q\times C}$: for an image $\boldsymbol{x}\in\mathbb{R}^{P\times Q\times C}$, $[\mathcal{F}(\boldsymbol{x})]_{i,j,c} = [\boldsymbol{x}]_{P+1-i,j,c}$ for $(i,j,c)\in[P]\times[Q]\times[C]$. Given a dataset $D = \{(\boldsymbol{x}_i, y_i)\}_{i=1}^N$, the *horizontal flip augmentation scheme* creates a new dataset of the form $D_{\mathcal{F}} = \{(\mathcal{F}(\boldsymbol{x}_i), y_i)\}_{i=1}^N$ and training is performed on $D_{\mathcal{F}} \cup D$.

## 4 Equivalence Between Augmented Kernel and Data Augmentation

In this section, we demonstrate the equivalence between using data augmentation and using a augmented kernel. To formally discuss the equivalence, we use group theory to describe translation and horizontal flip operators. We provide the definition of group in Section B for completeness.

It is easy to verify that $\{\mathcal{F}, \mathcal{I}\}$, $\{\mathcal{T}_{i,j}\}_{(i,j)\in[P]\times[Q]}$, $\{\mathcal{T}_{i,j}\circ\mathcal{F}\}_{(i,j)\in[P]\times[Q]} \cup \{\mathcal{T}_{i,j}\}_{(i,j)\in[P]\times[Q]}$ are groups, where $\mathcal{I}$ is the identity map. From now on, given a dataset $(\mathbf{X}, \boldsymbol{y})$ with data $\{(\boldsymbol{x}_i, y_i)\}_{i=1}^N$ and a group $\mathcal{G}$, the augmented dataset $(\mathbf{X}_{\mathcal{G}}, \boldsymbol{y}_{\mathcal{G}})$ is defined to be $\{g(\boldsymbol{x}_i), y_i\}_{g\in\mathcal{G}, i\in[N]}$. For kernel prediction for unseen data $\boldsymbol{x}'$ on the augmented dataset, we have the following formula: $\sum_{i\in[N], g\in\mathcal{G}} \widetilde{\alpha}_{i,g}\mathbf{K}(\boldsymbol{x}', g(\boldsymbol{x}_i))$, where $\widetilde{\boldsymbol{\alpha}} = \mathbf{K}_{\mathbf{X}_{\mathcal{G}}}^{-1}\boldsymbol{y}_{\mathcal{G}}$.

To proceed, we define the concept of *augmented kernel*. Let $\mathcal{G}$ be a finite group. Define the augmented kernel $\mathbf{K}^{\mathcal{G}}$ as $\mathbf{K}^{\mathcal{G}}(\boldsymbol{x}, \boldsymbol{x}') = \mathbb{E}_{g\in\mathcal{G}}\mathbb{E}_{g'\in\mathcal{G}}\mathbf{K}(g(\boldsymbol{x}), g'(\boldsymbol{x}'))$ where $\boldsymbol{x}, \boldsymbol{x}'$ are two inputs images. A key observation is that for CNTK and CNN-GP, when circular padding and GAP is adopted, these are actually the augmented kernels with the group $\mathcal{G} = \{\mathcal{T}_{i,j}\}_{(i,j)\in[P]\times[Q]}$. Formally, we have $\mathbf{\Sigma}_{\mathsf{GAP}}(\boldsymbol{x}, \boldsymbol{x}') = \frac{1}{PQ}\mathbf{\Sigma}_{\mathsf{FC}}^{\mathcal{G}}(\boldsymbol{x}, \boldsymbol{x}')$ and $\mathbf{\Theta}_{\mathsf{GAP}}(\boldsymbol{x}, \boldsymbol{x}') = \frac{1}{PQ}\mathbf{\Theta}_{\mathsf{FC}}^{\mathcal{G}}(\boldsymbol{x}, \boldsymbol{x}')$. The proof for these two equations is just by checking the formula of these kernels and using definition of circular padding. By similar proof, one can observe the following invariance property of $\mathbf{\Sigma}_{\mathsf{GAP}}, \mathbf{\Sigma}_{\mathsf{FC}}, \mathbf{\Theta}_{\mathsf{GAP}}$ and $\mathbf{\Theta}_{\mathsf{FC}}$, under all groups mentioned above, including $\{\mathcal{F}, \mathcal{I}\}$ and $\{\mathcal{T}_{i,j}\}_{(i,j)\in[P]\times[Q]}$.

**Definition 4.1.** *A kernel* $\mathbf{K}$ *is* invariant *under a group* $\mathcal{G}$ *if and only if for any* $g \in \mathcal{G}$, $\mathbf{K}(g(\boldsymbol{x}), g(\boldsymbol{x}')) = \mathbf{K}(\boldsymbol{x}, \boldsymbol{x}')$.

Now the following theorem formally states the equivalence between using an augmented kernel on the dataset and using the kernel on the augmented dataset.

**Theorem 4.1.** *Given a group $\mathcal{G}$ and a kernel $\mathbf{K}$ such that $\mathbf{K}$ is invariant under $\mathcal{G}$, then the prediction of augmented kernel $\mathbf{K}^{\mathcal{G}}$ with dataset $(\mathbf{X}, \boldsymbol{y})$ is equal to that of kernel $\mathbf{K}$ and augmented dataset $(\mathbf{X}_{\mathcal{G}}, \boldsymbol{y}_{\mathcal{G}})$. Namely, for any $\boldsymbol{x}' \in \mathbb{R}^{P \times Q \times C}$, $\sum_{i=1}^{N} \alpha_i \mathbf{K}^{\mathcal{G}}(\boldsymbol{x}', \boldsymbol{x}_i) = \sum_{i \in [N], g \in \mathcal{G}} \widetilde{\alpha}_{i,g} \mathbf{K}(\boldsymbol{x}', g(\boldsymbol{x}_i))$ where $\boldsymbol{\alpha} = \left(\mathbf{K}_{\mathbf{X}}^{\mathcal{G}}\right)^{-1} \boldsymbol{y}, \widetilde{\boldsymbol{\alpha}} = \left(\mathbf{K}_{\mathbf{X}_{\mathcal{G}}}\right)^{-1} \boldsymbol{y}_{\mathcal{G}}$.*

The proof is deferred to Appendix B. Two corollaries are directly followed.

**Corollary 4.1.** *For $\mathcal{G} = \{\mathcal{T}_{i,j}\}_{(i,j) \in [P] \times [Q]}$, for any given dataset $D$, the prediction of $\boldsymbol{\Sigma}_{GAP}$ (or $\boldsymbol{\Theta}_{GAP}$) with dataset $D$ is equal to the prediction of $\boldsymbol{\Sigma}_{FC}$ (or $\boldsymbol{\Theta}_{FC}$) with augmented dataset $D_{\mathcal{T}}$.*

**Corollary 4.2.** *For $\mathcal{G} = \{\mathcal{F}, \mathcal{I}\}$, for any given dataset $D$, the prediction of $\boldsymbol{\Sigma}_{GAP}^{\mathcal{G}}$ (or $\boldsymbol{\Theta}_{GAP}^{\mathcal{G}}$) with dataset $D$ is equal to the prediction of $\boldsymbol{\Sigma}_{GAP}$ (or $\boldsymbol{\Theta}_{GAP}$) with augmented dataset $D_{\mathcal{F}} \cup D$.*

Now we discuss implications of Theorem 4.1 and its corollaries. Naively applying data augmentation, with full translation on CNTK or CNN-GP for example, one needs to create a $P^2 Q^2$ times larger kernel matrix since there are $PQ$ translation operators, which is often computationally infeasible. Instead, we can directly use the augmented kernel ($\boldsymbol{\Sigma}_{GAP}$ or $\boldsymbol{\Theta}_{GAP}$ for the case of full translation on CNTK or CNN-GP) for prediction, for which one only needs to create a kernel matrix that is as large as the original one. For horizontal flip, although the augmentation kernel is not as conveniently computed as full translation, Corollary 4.2 still provides a more efficient method for computing kernel value and solving kernel regression, since the augmented dataset is twice as large as the original dataset.

## 5  LOCAL AVERAGE POOLING

In this section, we introduce a new operation called *Local Average Pooling* (LAP). As discussed in the introduction, full translation data augmentation can create unrealistic images. A natural idea is to do local translation data augmentation, i.e., restricting the distance of translation. More specifically, we only allow translation operations $\mathcal{T}_{\Delta_i, \Delta_j}$ (cf. Section 3.3) for $(\Delta_i, \Delta_j) \in [-c, c] \times [-c, c]$ where $c$ is a parameter to control the amount of allowed translation. With a proper choice of the parameter $c$, translation data augmentation will not create unrealistic images (cf. Figure 2). However, naive local translation data augmentation is computationally infeasible for kernel methods, even for moderate choice of $c$. To remedy this issue, in this section we introduce LAP, which is inspired by the connection between full translation data augmentation and GAP on CNN-GP and CNTK. Here, for simplicity, we assume $P = Q$ and derive the formula only for CNTK. Our formula can be generalized to CNN-GP in a straightforward manner.

Recall that for two given images $\boldsymbol{x}$ and $\boldsymbol{x}'$, without GAP, the formula for output of CNTK is $\mathrm{tr}\left(\boldsymbol{\Theta}(\boldsymbol{x}, \boldsymbol{x}')\right)$. With GAP, the formula for output of CNTK is $\frac{1}{P^4} \sum_{i,j,i',j' \in [P]^4} \left[\boldsymbol{\Theta}\left(\boldsymbol{x}, \boldsymbol{x}'\right)\right]_{i,j,i',j'}$. With circular padding, the formula can be rewritten as $\frac{1}{P^2} \mathbb{E}_{\Delta_i, \Delta_i', \Delta_j, \Delta_j' \sim [P]^4} \sum_{i,j \in [P] \times [P]} \left[\boldsymbol{\Theta}\left(\boldsymbol{x}, \boldsymbol{x}'\right)\right]_{i+\Delta_i, j+\Delta_j, i+\Delta_i', j+\Delta_j'}$, which is again equal to $\frac{1}{P^2} \mathbb{E}_{\Delta_i, \Delta_i', \Delta_j, \Delta_j' \sim [P]^4} \mathrm{tr}\left(\boldsymbol{\Theta}\left(\mathcal{T}_{\Delta_i, \Delta_j}(\boldsymbol{x}), \mathcal{T}_{\Delta_i', \Delta_j'}(\boldsymbol{x}')\right)\right)$. We ignore the $1/P^2$ scaling factor since it plays no role in kernel regression.

Now we consider restricted translation operations $\mathcal{T}_{\Delta_i, \Delta_j}$ with $(\Delta_i, \Delta_j) \in [-c, c] \times [-c, c]$ and derive the formula for LAP. Assuming circular padding, we have

$$\mathbb{E}_{\Delta_i, \Delta_i', \Delta_j, \Delta_j' \sim [-c,c]^4} \mathrm{tr}\left(\boldsymbol{\Theta}\left(\mathcal{T}_{\Delta_i, \Delta_j}(\boldsymbol{x}), \mathcal{T}_{\Delta_i', \Delta_j'}(\boldsymbol{x}')\right)\right)$$
$$= \frac{1}{(2c+1)^4} \sum_{\Delta_i, \Delta_i', \Delta_j, \Delta_j' \in [-c,c]^4} \sum_{i,j \in [P]^2} \left[\boldsymbol{\Theta}(\boldsymbol{x}, \boldsymbol{x}')\right]_{i+\Delta_i, j+\Delta_j, i+\Delta_i', j+\Delta_j'}. \tag{1}$$

Now we have derived the formula for LAP which is given in Equation 1. Notice that the formula in Equation 1 is a well-defined quantity for all padding schemes. In particular, assuming zero padding, when $c = P$, LAP is equivalent to GAP. When $c = 0$, LAP is equivalent to no pooling layer. Another advantage of LAP is that it does not incur significant additional computational cost, since the formula in Equation 1 can be rewritten as $\sum_{i,j,i',j' \in [P]^4} [\boldsymbol{w}]_{i,j,i',j'} \cdot \left[\boldsymbol{\Theta}(\boldsymbol{x}, \boldsymbol{x}')\right]_{i,j,i',j'}$ where each entry in the weight tensor $\boldsymbol{w}$ can be calculated in $O(1)$ time.

Note that the GAP operation in CNN-GP and CNTK corresponds to the GAP layer in CNNs. Here we observe the following *box filtering layer* that corresponds to LAP in CNNs. Box filtering layer

(BF) is a function $\mathbb{R}^{P \times Q} \to \mathbb{R}^{P \times Q}$ such that $[\mathsf{BF}(\boldsymbol{x})]_{i,j} = \frac{1}{(2c+1)^2} \sum_{\Delta_i, \Delta_j \in [-c,c]^2} \boldsymbol{x}_{i+\Delta_i, j+\Delta_j}$. This is in fact a standard average pooling layer but with stride 1 and pooling size $2c+1$. We prove the equivalence between LAP and box filtering layer in Appendix C. In Section 6.3, we test BF on CNNs to verify its effectiveness.

# 6 EXPERIMENTS

In this section we present our empirical findings on CIFAR-10 (Krizhevsky, 2009) and Fashion-MNIST (Xiao et al., 2017). The detailed experimental setup is reported in Appendix D. When reporting test accuracies, the best result on the test set is in boldface and the result that corresponds to the hyper-parameter chosen by cross-validation is underlined.

## 6.1 ABLATION STUDY ON CIFAR-10 AND FASHION-MNIST

We perform experiments to study the effect of different values of the $c$ parameter in LAP and horizontal flip data argumentation on CNTK and CNN-GP. For experiments in this section we set the bias term in CNTK and CNN-GP to be $\beta = 0$ (cf. Section A). We use the same architecture for CNTK and CNN-GP as in Arora et al. (2019). I.e., we stack multiple convolutional layers before the final pooling layer. We use $d$ to denote the number of convolutions layers, and in our experiments we set $d$ to be 5, 8, 11 or 14, to study the effect of depth on CNTK and CNN-GP. For CIFAR-10, we set the $c$ parameter in LAP to be $0, 4, \ldots, 32$, while for Fashion-MNIST we set the $c$ parameter in LAP to be $0, 4, \ldots, 28$. Notice that when $c = 32$ for CIFAR-10 or $c = 28$ for Fashion-MNIST, LAP is equivalent to GAP, and when $c = 0$, LAP is equivalent to no pooling layer. Results on CIFAR-10 are reported in Tables 1 and 3. Due to space constraint, results on Fashion-MNIST are reported in Tables 5 and 6 in Appendix E. In each table, for each combination of $c$ and $d$, the first number is the test accuracy without horizontal flip data augmentation (in percentage), and the second number (in parentheses) is the test accuracy with horizontal flip data augmentation. To perform cross-validation to choose the hyper-parameters, we use the last 10000 samples in the training set of CIFAR-10 and Fashion-MNIST as the validation set and the rest samples as the training set. We then use the full training set to report the test accuracy. To perform cross-validation, we choose $c$, $d$, CNN or CNN-GP, and whether or not to adopt horizontal flip based on the validation accuracy (shown in Appendix F). With cross-validation, the resulting accuracy is 82.09% on CIFAR-10 and 94.07% on Fashion-MNIST.

We made the following observations regarding our experimental results.

- LAP with a proper choice of the parameter $c$ significantly improves the performance of CNTK and CNN-GP. On CIFAR-10, the best-performing value of $c$ is $c = 12$ or $16$, while on Fashion-MNIST the best-performing value of $c$ is $c = 4$. We suspect this difference is due to the nature of the two datasets: CIFAR-10 contains real-life images and thus allow more translation, while Fashion-MNIST contains images with centered clothes and thus allow less translation. For both datasets, the best-performing value of $c$ is consistent across all settings (depth, CNTK or CNN-GP) that we have considered.
- Horizontal flip data augmentation is less effective on Fashion-MNIST than on CIFAR-10. There are two possible explanations for this phenomenon. First, most images in Fashion-MNIST are nearly horizontally symmetric (e.g., T-shirts and bags). Second, CNTK and CNN-GP have already achieved a relatively high accuracy on Fashion-MNIST, and thus it is reasonable for horizontal flip data augmentation to be less effective on this dataset.
- Finally, for CNTK, when $c = 0$ (no pooling layer) and $c = 32$ (GAP) our reported test accuracies are close to those in Arora et al. (2019) on CIFAR-10. For CNN-GP, when $c = 0$ (no pooling layer) our reported test accuracies are close to those in Novak et al. (2019) on CIFAR-10 and Fashion-MNIST. This suggests that we have reproduced previous reported results.

## 6.2 IMPROVING PERFORMANCE ON CIFAR-10 USING RANDOM PATCHES LAYER

Finally, we explore another interesting question: what is the best performance achievable via a method that is not a trained neural network? To further improve the performance, we combine CNTK and CNN-GP with LAP, together with the unsupervised learning approach developed in Coates et al. (2011). Here we use the variant implemented in Recht et al. (2019). More specifically, we first sample *2048* random image patches with size $5 \times 5$ from all training images. Then for the sampled images patches, we subtract the mean of the patches, then normalize them to have unit

| $c$ \\ $d$ | 5 | 8 | 11 | 14 |
|---|---|---|---|---|
| 0 | 66.55 (69.87) | 66.27 (69.87) | 65.85 (69.37) | 65.47 (68.90) |
| 4 | 77.06 (79.08) | 77.14 (78.96) | 77.06 (78.98) | 76.52 (78.74) |
| 8 | 79.24 (80.95) | 79.25 (81.03) | 78.98 (80.94) | 78.65 (80.35) |
| 12 | **80.11** (81.34) | 79.79 (81.28) | 79.29 (81.14) | 79.13 (80.91) |
| 16 | 79.80 (81.21) | 79.71 (**81.40**) | 79.74 (81.09) | 79.42 (81.00) |
| 20 | 79.24 (80.67) | 79.27 (80.88) | 79.30 (80.76) | 78.92 (80.39) |
| 24 | 78.07 (79.88) | 78.16 (79.79) | 78.14 (80.06) | 77.87 (80.07) |
| 28 | 76.91 (78.69) | 77.33 (79.20) | 77.65 (79.56) | 77.65 (79.74) |
| 32 | 76.79 (78.53) | 77.39 (79.13) | 77.63 (79.51) | 77.63 (79.74) |

Table 1: Test accuracy of CNTK on CIFAR-10.

| $c$ \\ $d$ | 5 | 8 | 11 | 14 |
|---|---|---|---|---|
| 4 | 84.63 (86.64) | 84.07 (86.23) | 83.29 (85.53) | 82.57 (84.81) |
| 8 | 86.36 (88.32) | 85.80 (87.81) | 85.01 (87.08) | 84.57 (86.53) |
| 12 | 86.74 (88.35) | 86.20 (87.90) | 85.60 (87.36) | 84.95 (86.99) |
| 16 | **86.77** (**88.36**) | 86.17 (87.85) | 85.60 (87.44) | 84.92 (86.98) |
| 20 | 86.17 (87.77) | 85.71 (87.50) | 85.14 (87.07) | 84.59 (86.84) |

Table 2: Test accuracy of random patches layer + CNTK on CIFAR-10.

norm, and finally perform ZCA transformation to the resulting patches. We use the resulting patches as 2048 filters of a convolutional layer with kernel size 5, stride 1 and no dilation or padding. For an input image $x$, we use $\text{conv}(x)$ to denote the output of the convolutional layer. As in the implementation in Recht et al. (2019), we use $\text{ReLU}(\text{conv}(x) - \beta_{\text{feature}})$ and $\text{ReLU}(-\text{conv}(x) - \beta_{\text{feature}})$ as the input feature for CNTK and CNN-GP. Here we fix $\beta_{\text{feature}} = 1$ as in Recht et al. (2019) and the bias term $\beta$ in CNTK and CNN-GP to be $\beta = 1$. To make the output kernel value invariant under horizontal flip (cf. Defintion 4.1), for each image patch, we horizontally flipped it and add the flipped patch into the convolutional layer as a new filter. Thus, for an input CIFAR-10 image of size $32 \times 32$, the dimension of the output feature is $8192 \times 28 \times 28$. To isolate the effect of randomness in the choices of the image patches, we fix the random seed to be 0 throughout the experiment. In this experiment, we set the value of the $c$ parameter in LAP to be $4, 8, 12, \dots, 20$ to avoid small and large values of $c$. The results are reported in Tables 2 and 4. Similar to the experiments in Section 6.1, again we set the hyper-parameters by cross-validation, and the resulting accuracy is 88.91%. See Appendix F for the validation accuracy for different hyper-parameters.

From our experimental results, it is evident that combining CNTK or CNN-GP with additional feature extractor can significantly improve upon the performance of using solely CNTK or CNN-GP, and that of using solely the feature extractor Coates et al. (2011). Previously, it has been reported in Recht et al. (2019) that using solely the feature extractor Coates et al. (2011) (together with appropriate pooling layer) can only achieve a test accuracy of 84.2% using *256, 000* image patches, or 83.3% using *32, 000* image patches. Even with the help of horizontal data augmentation, the feature extractor Coates et al. (2011) can only achieve a test accuracy of 85.6% using *256, 000* image patches, or 85.0% using *32, 000* image patches. Here we use significantly less image patches (only *2048*) but achieve a much better performance, with the help of CNTK and CNN-GP. In particular, we achieve a performance of 88.91% on CIFAR-10, matching the performance of AlexNet on the same dataset. In the setting reported in Coates et al. (2011), increasing the number of sampled image patches will further improve the performance. Here we also conjecture that in our setting, further increasing the number of sampled image patches can improve the performance and get close to modern CNNs. However, due the limitation on computational resources, we leave exploring the effect of number of sampled image patches as a future research direction.

## 6.3 EXPERIMENTS ON CNN WITH BOX FILTERING LAYER

In Figure 1, we verify the effectiveness of BF on a 10-layer CNN (with Batch Normalization) on CIFAR-10. The setting of this experiment is reported in Appendix G. Our network structure has no pooling layer except for the BF layer before the last fully-connected layer. The fully-connected layer is fixed during the training. Our experiment illustrates that even with a fixed last FC layer, using

| $c$ \ $d$ | 5 | 8 | 11 | 14 |
|---|---|---|---|---|
| 0 | 63.53 (67.90) | 65.54 (69.43) | 66.42 (70.30) | 66.81 (70.48) |
| 4 | 76.35 (78.79) | 77.03 (79.30) | 77.39 (79.52) | 77.35 (79.65) |
| 8 | 79.48 (81.32) | 79.82 (81.49) | 79.76 (81.71) | 79.69 (81.53) |
| 12 | 80.40 (82.13) | 80.64 (82.09) | 80.58 (82.06) | 80.32 (81.95) |
| 16 | 80.36 (81.73) | **80.78** (**82.20**) | 80.59 (82.06) | 80.41 (81.83) |
| 20 | 79.87 (81.50) | 80.15 (81.33) | 79.87 (81.46) | 79.98 (81.35) |
| 24 | 78.60 (79.98) | 78.91 (80.48) | 79.22 (80.53) | 78.94 (80.46) |
| 28 | 77.18 (78.84) | 78.03 (79.86) | 78.45 (79.87) | 78.48 (80.07) |
| 32 | 77.00 (78.49) | 77.85 (79.65) | 78.49 (80.04) | 78.45 (80.01) |

Table 3: Test accuracy of CNN-GP on CIFAR-10.

| $c$ \ $d$ | 5 | 8 | 11 | 14 |
|---|---|---|---|---|
| 4 | 85.49 (87.32) | 85.37 (87.22) | 85.16 (87.11) | 84.79 (86.81) |
| 8 | 87.07 (88.64) | 86.82 (88.68) | 86.53 (88.40) | 86.39 (88.15) |
| 12 | 87.23 (88.91) | 87.12 (**88.92**) | 86.87 (88.66) | 86.62 (88.29) |
| 16 | **87.28** (88.90) | 87.11 (88.66) | 86.92 (88.61) | 86.74 (88.24) |
| 20 | 86.81 (88.26) | 86.77 (88.24) | 86.61 (88.14) | 86.26 (87.84) |

Table 4: Test accuracy of random patches layer + CNN-GP on CIFAR-10.

GAP could improve the performance of CNN. Our experiments also show that BF with appropriate choice of $c$ achieves better performance than GAP.

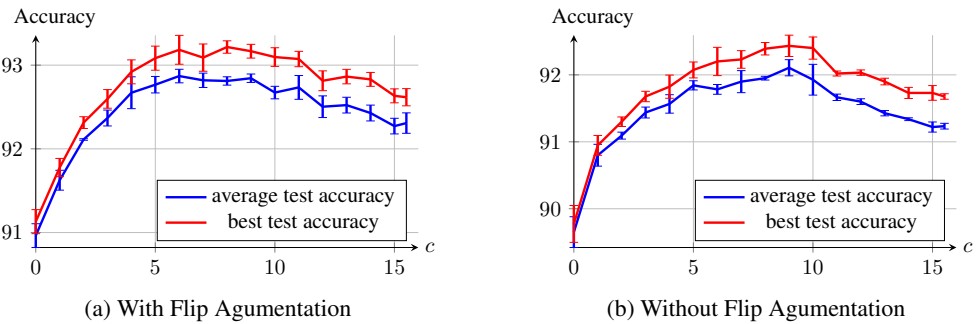

(a) With Flip Agumentation    (b) Without Flip Agumentation

Figure 1: Test accuracy of 10-layer CNN with various values for the $c$ parameter in BF.

## 7 CONCLUSION

In this paper, inspired by the connection between full translation data augmentation and GAP, we derive a new operation, LAP, on CNTK and CNN-GP, which consistently improves the performance on image classification tasks. Combining CNN-GP with LAP and the pre-processing technique proposed by Coates et al. (2011), the resulting kernel achieves 89% accuracy on CIFAR-10, matching the performance of AlexNet and is the strongest classifier that is not a trained neural network.

Here we list a few future research directions. Is it possible to develop analogs of CNTK or CNN-GP incorporating modern techniques such as batch norm and residual layers, to further improve the performance? Moreover, it is an interesting direction to study other components in modern CNNs through the lens of CNTK and CNN-GP.

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

## A    FORMAL DEFINITIONS OF CNN-GP AND CNTK

We add some additional notations. Let $\boldsymbol{I}$ be the identity matrix, and $[n] = \{1, 2, \ldots, n\}$. Let $\boldsymbol{e}_i$ be an indicator vector with $i$-th entry being 1 and other entries being 0, and let $\mathbf{1}$ denote the all-one vector. We use $\odot$ to denote the pointwise product and $\otimes$ to denote the tensor product. We use $\mathrm{diag}(\cdot)$ to transform a vector to a diagonal matrix. We use $\sigma(\cdot)$ to denote the activation function, such as the rectified linear unit (ReLU) function: $\sigma(z) = \max\{z, 0\}$, and $\dot{\sigma}(\cdot)$ to denote the derivative of $\sigma(\cdot)$. We set $c_\sigma = 2$. Denote by $\mathcal{N}(\boldsymbol{\mu}, \boldsymbol{\Sigma})$ the Gaussian distribution with mean $\boldsymbol{\mu}$ and covariance $\boldsymbol{\Sigma}$.

Equation equation 2 shows patch $[\boldsymbol{w} * \boldsymbol{x}]_{ij}$ depends on $[\boldsymbol{x}]_{i-\frac{q-1}{2}:i+\frac{q-1}{2}, j-\frac{q-1}{2}:j+\frac{q-1}{2}}$. For $(i, j, i', j') \in [P] \times [Q] \times [P] \times [Q]$, define

$$\mathcal{D}_{ij,i'j'} = \{(i+a, j+b, i'+a', j'+b') \in [P] \times [Q] \times [P] \times [Q] \mid -(q-1)/2 \leq a, b, a', b' \leq (q-1)/2\}.$$

Now we define the convolution operation. For a convolutional filter $\boldsymbol{w} \in \mathbb{R}^{q \times q}$ and an image $\boldsymbol{x} \in \mathbb{R}^{P \times Q}$, the convolution operator is defined as

$$[\boldsymbol{w} * \boldsymbol{x}]_{ij} = \sum_{a=-\frac{q-1}{2}}^{\frac{q-1}{2}} \sum_{b=-\frac{q-1}{2}}^{\frac{q-1}{2}} [\boldsymbol{w}]_{a+\frac{q+1}{2}, b+\frac{q+1}{2}} [\boldsymbol{x}]_{a+i, b+j} \text{ for } i \in [P], j \in [Q]. \tag{2}$$

Now we formally define CNN.

- Let $\boldsymbol{x}^{(0)} = \boldsymbol{x} \in \mathbb{R}^{P \times Q \times C^{(0)}}$ be the input image where $C^{(0)}$ is the initial number of channels.
- For $h = 1, \ldots, L, \beta = 1, \ldots, C^{(h)}$, the intermediate outputs are defined as

$$\tilde{\boldsymbol{x}}_{(\beta)}^{(h)} = \sum_{\alpha=1}^{C^{(h-1)}} \boldsymbol{W}_{(\alpha),(\beta)}^{(h)} * \boldsymbol{x}_{(\alpha)}^{(h-1)} + \gamma \cdot b_{(\beta)}, \quad \boldsymbol{x}_{(\beta)}^{(h)} = \sqrt{\frac{c_\sigma}{C^{(h)} \times q \times q}} \sigma\left(\tilde{\boldsymbol{x}}_{(\beta)}^{(h)}\right)$$

where each $\boldsymbol{W}_{(\alpha),(\beta)}^{(h)} \in \mathbb{R}^{q \times q}$ is a filter with Gaussian initialization and $b_{(\beta)}$ is a bias term with Gaussian initialization scaled by $\gamma$.

**CNN-GP and CNTK**

- For $\alpha = 1, \ldots, C^{(0)}, (i, j, i', j') \in [P] \times [Q] \times [P] \times [Q]$, define

$$\boldsymbol{K}_{(\alpha)}^{(0)}(\boldsymbol{x}, \boldsymbol{x}') = \boldsymbol{x}_{(\alpha)} \otimes \boldsymbol{x}'_{(\alpha)} \text{ and } \left[\boldsymbol{\Sigma}^{(0)}(\boldsymbol{x}, \boldsymbol{x}')\right]_{ij,i'j'} = \frac{1}{q^2} \sum_{\alpha=1}^{C^{(0)}} \mathrm{tr}\left(\left[\boldsymbol{K}_{(\alpha)}^{(0)}(\boldsymbol{x}, \boldsymbol{x}')\right]_{\mathcal{D}_{ij,i'j'}}\right) + \beta^2.$$

- For $h \in [L]$,
  - For $(i, j, i', j') \in [P] \times [Q] \times [P] \times [Q]$, define

$$\boldsymbol{\Lambda}_{ij,i'j'}^{(h)}(\boldsymbol{x}, \boldsymbol{x}') = \begin{pmatrix} \left[\boldsymbol{\Sigma}^{(h-1)}(\boldsymbol{x}, \boldsymbol{x})\right]_{ij,ij} & \left[\boldsymbol{\Sigma}^{(h-1)}(\boldsymbol{x}, \boldsymbol{x}')\right]_{ij,i'j'} \\ \left[\boldsymbol{\Sigma}^{(h-1)}(\boldsymbol{x}', \boldsymbol{x})\right]_{i'j',ij} & \left[\boldsymbol{\Sigma}^{(h-1)}(\boldsymbol{x}', \boldsymbol{x}')\right]_{i'j',i'j'} \end{pmatrix} \in \mathbb{R}^{2 \times 2}.$$

  - Define $\boldsymbol{K}^{(h)}(\boldsymbol{x}, \boldsymbol{x}'), \dot{\boldsymbol{K}}^{(h)}(\boldsymbol{x}, \boldsymbol{x}') \in \mathbb{R}^{P \times Q \times P \times Q}$, for $(i, j, i', j') \in [P] \times [Q] \times [P] \times [Q]$

$$\left[\boldsymbol{K}^{(h)}(\boldsymbol{x}, \boldsymbol{x}')\right]_{ij,i'j'} = c_\sigma \cdot \mathop{\mathbb{E}}_{(u,v) \sim \mathcal{N}\left(\mathbf{0}, \boldsymbol{\Lambda}_{ij,i'j'}^{(h)}(\boldsymbol{x}, \boldsymbol{x}')\right)} [\cdot \sigma(u) \sigma(v)], \tag{3}$$

$$\left[\dot{\boldsymbol{K}}^{(h)}(\boldsymbol{x}, \boldsymbol{x}')\right]_{ij,i'j'} = c_\sigma \cdot \mathop{\mathbb{E}}_{(u,v) \sim \mathcal{N}\left(\mathbf{0}, \boldsymbol{\Lambda}_{ij,i'j'}^{(h)}(\boldsymbol{x}, \boldsymbol{x}')\right)} [\dot{\sigma}(u) \dot{\sigma}(v)]. \tag{4}$$

  - Define $\boldsymbol{\Sigma}^{(h)}(\boldsymbol{x}, \boldsymbol{x}') \in \mathbb{R}^{P \times Q \times P \times Q}$, for $(i, j, i', j') \in [P] \times [Q] \times [P] \times [Q]$

$$\left[\boldsymbol{\Sigma}^{(h)}(\boldsymbol{x}, \boldsymbol{x}')\right]_{ij,i'j'} = \frac{1}{q^2} \mathrm{tr}\left(\left[\boldsymbol{K}^{(h)}(\boldsymbol{x}, \boldsymbol{x}')\right]_{D_{ij,i'j'}}\right) + \beta^2.$$

Note that $\boldsymbol{\Sigma}(\boldsymbol{x}, \boldsymbol{x}')$ and $\dot{\boldsymbol{\Sigma}}(\boldsymbol{x}, \boldsymbol{x}')$ share similar structures as their NTK counterparts (Jacot et al., 2018). The only difference is that we have one more step, taking the trace over patches. This step represents the convolution operation in the corresponding CNN. Next, we can use a recursion to compute the final kernel value.

1. First, we define $\mathbf{\Theta}^{(0)}(\boldsymbol{x}, \boldsymbol{x}') = \mathbf{\Sigma}^{(0)}(\boldsymbol{x}, \boldsymbol{x}')$.
2. For $h = 1, \ldots, L$ and $(i, j, i', j') \in [P] \times [Q] \times [P] \times [Q]$, we define

$$
\left[\mathbf{\Theta}^{(h)}(\boldsymbol{x}, \boldsymbol{x}')\right]_{ij,i'j'} = \frac{1}{q^2} \mathrm{tr}\left(\left[\dot{\boldsymbol{K}}^{(h)}(\boldsymbol{x}, \boldsymbol{x}') \odot \mathbf{\Theta}^{(h-1)}(\boldsymbol{x}, \boldsymbol{x}') + \boldsymbol{K}^{(h)}(\boldsymbol{x}, \boldsymbol{x}')\right]_{D_{ij,i'j'}}\right) + \beta^2.
$$

# B    ADDITIONAL DEFINITION AND PROOF FOR SECTION 4

**Definition B.1** (Group). $(\mathcal{G}, \circ)$ *is a* group, *if and only if*

1. *each element $g \in \mathcal{G}$ is a operator:* $\mathbb{R}^{P \times Q \times C} \to \mathbb{R}^{P \times Q \times C}$;
2. $\forall g_1, g_2 \in \mathcal{G}, g_1 \circ g_2 \in \mathcal{G}$, *where* $(g_1 \circ g_2)(\boldsymbol{x})$ *is defined as* $g_1(g_2(\boldsymbol{x}))$.
3. $\forall g_1, g_2, g_3 \in \mathcal{G}, (g_1 \circ g_2) \circ g_3 = g_1 \circ (g_2 \circ g_3)$.
4. $\exists e \in \mathcal{G}$, *such that* $\forall g \in \mathcal{G}, e \circ g = g \circ e = g$.
5. $\forall g_1 \in \mathcal{G}, \exists g_2 \in \mathcal{G}$, *such that* $g_1 \circ g_2 = g_2 \circ g_1 = e$. *We denote $g_2$ as the inverse of $g_1$, namely,* $g_1^{-1}$.

*Proof of Theorem 4.1.* Since we assume $\mathbf{K}_{\mathbf{X}}^{\mathcal{G}}$ and $\mathbf{K}_{\mathbf{X}_{\mathcal{G}}}$ are invertible, both $\boldsymbol{\alpha}$ and $\widetilde{\boldsymbol{\alpha}}$ are uniquely defined. Now we claim $\widetilde{\boldsymbol{\alpha}}_g = \{\widetilde{\alpha}_{i,g}\}_{i \in [N]} \in \mathbb{R}^N$ is equal to $\frac{\boldsymbol{\alpha}}{|\mathcal{G}|}$ for all $g \in \mathcal{G}$.

By the invariance of $\mathbf{K}$ under $\mathcal{G}$, for all $j \in [N]$ and $g' \in \mathcal{G}$,

$$
\sum_{i \in [N], g \in \mathcal{G}} \frac{\alpha_i}{|\mathcal{G}|} \mathbf{K}(g'(\boldsymbol{x}_j), g(\boldsymbol{x}_i)) = \sum_{i \in [N], g \in \mathcal{G}} \frac{\alpha_i}{|\mathcal{G}|} \mathbf{K}((g^{-1} \circ g')(\boldsymbol{x}_j), \boldsymbol{x}_i)
$$

$$
= \sum_{i \in [N]} \alpha_i \mathbb{E}_{g \in \mathcal{G}} \mathbf{K}(g(\boldsymbol{x}_j), \boldsymbol{x}_i)
$$

$$
= \sum_{i \in [N]} \alpha_i \mathbf{K}^{\mathcal{G}}(\boldsymbol{x}_j, \boldsymbol{x}_i)
$$

$$
= y_j.
$$

Note that $\widetilde{\boldsymbol{\alpha}}$ is defined as the unique solution of $\mathbf{K}_{\mathbf{X}_{\mathcal{G}}} \widetilde{\boldsymbol{\alpha}} = \boldsymbol{y}_{\mathcal{G}}$, the claim has been verified.

Similarly, we have

$$
\sum_{i \in [N], g \in \mathcal{G}} \frac{\alpha_i}{|\mathcal{G}|} \mathbf{K}(\boldsymbol{x}', g(\boldsymbol{x}_i)) = \sum_{i \in [N]} \alpha_i \mathbb{E}_{g \in \mathcal{G}} \mathbf{K}(g^{-1}(\boldsymbol{x}'), \boldsymbol{x}_i) = \sum_{i \in [N]} \alpha_i \mathbf{K}^{\mathcal{G}}(\boldsymbol{x}', \boldsymbol{x}_i).
$$

$\square$

# C    EQUIVALENCE BETWEEN LAP AND BOX FILTERING LAYER.

For a CNN with a box filtering layer before the final fully-connected layer, the final output is defined as $f(\boldsymbol{\theta}, \boldsymbol{x}) = \sum_{\alpha=1}^{C^{(L)}} \left\langle \boldsymbol{W}_{(\alpha)}^{(L+1)}, \mathsf{BF}\left(\boldsymbol{x}_{(\alpha)}^{(L)}\right) \right\rangle$, where $\boldsymbol{x}_{(\alpha)}^{(L)} \in \mathbb{R}^{P \times Q}$, and $\boldsymbol{W}_{(\alpha)}^{(L+1)} \in \mathbb{R}^{P \times Q}$ is the weight of the last fully-connected layer.

Now we establish the equivalence between BF and LAP on CNTK. The equivalence on CNN-GP can be derived similarly. Let $\mathbf{\Theta}_{\mathsf{BF}}(\boldsymbol{x}, \boldsymbol{x}') \in \mathbb{R}^{[P] \times [Q] \times [P] \times [Q]}$ be the CNTK kernel of $\mathsf{BF}\left(\boldsymbol{x}_{(\alpha)}^{(L)}\right)$. Since BF is just a linear operation, we have

$$
[\mathbf{\Theta}_{\mathsf{BF}}(\boldsymbol{x}, \boldsymbol{x}')]_{i,j,i',j'} = \frac{1}{(2c+1)^4} \sum_{\Delta_i, \Delta_j, \Delta'_i, \Delta'_j \in [-c,c]^4} \left[\mathbf{\Theta}^{(L)}(\boldsymbol{x}, \boldsymbol{x}')\right]_{i+\Delta_i, j+\Delta_j, i'+\Delta'_i, j'+\Delta'_j}.
$$

By the formula of the output kernel value for CNTK without GAP, we obtain

$$
\mathrm{tr}\left(\mathbf{\Theta}_{\mathsf{BF}}(\boldsymbol{x}, \boldsymbol{x}')\right) = \frac{1}{(2c+1)^4} \sum_{\Delta_i, \Delta'_i, \Delta_j, \Delta'_j \in [-c,c]^4} \sum_{i,j \in [P] \times [Q]} [\mathbf{\Theta}(\boldsymbol{x}, \boldsymbol{x}')]_{i+\Delta_i, j+\Delta_j, i+\Delta'_i, j+\Delta'_j}.
$$

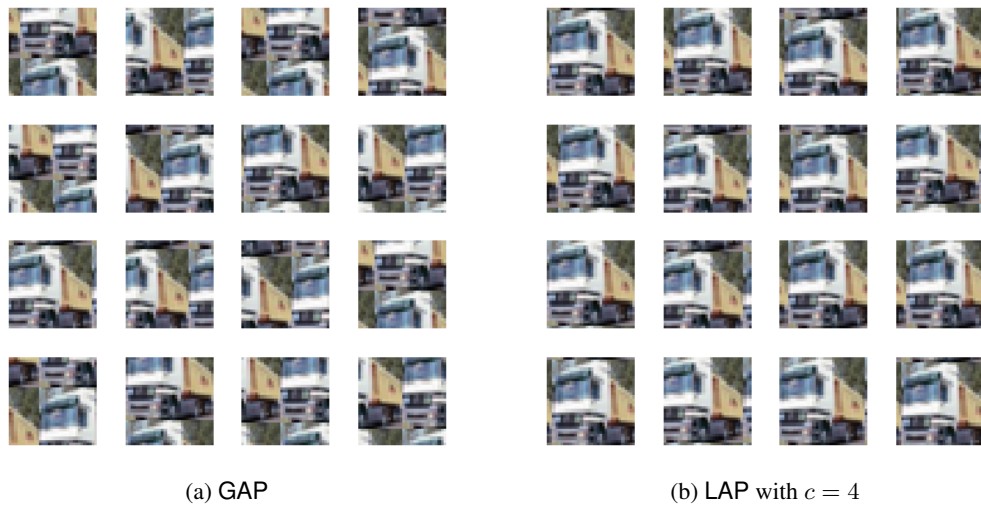

(a) GAP  (b) LAP with $c = 4$

Figure 2: Randomly sampled images with full translation data augmentation and local translation data augmentation from CIFAR-10. Full translation data augmentation can create unrealistic images that harm the performance whereas local translation data augmentation creates more realistic images.

# D   EXPERIMENTAL SETUP IN SECTION 6

For both CIFAR-10 and Fashion-MNIST we use the full training set and report the test accuracy on the full test set. Throughout this section we only consider $3 \times 3$ convolutional filters with stride 1 and no dilation. In the convolutional layers in CNTK and CNN-GP, we use zero padding with pad size 1 to ensure the input of each layer has the same size. We use zero padding for LAP throughout the experiment. We perform standard preprocessing (mean subtraction and standard deviation division) for all images.

In all experiments, we perform kernel ridge regression to utilize the calculated kernel values[4]. We normalize the kernel matrices so that all diagonal entries are ones. Equivalently, we ensure all features have unit norm in RKHS. Since the resulting kernel matrices are usually ill-conditioned, we set the regularization term $\lambda = 5 \times 10^{-5}$, to make inverting kernel matrices numerically stable. We use one-hot encodings of the labels as regression targets. We use scipy.linalg.solve to solve the corresponding kernel ridge regression problem.

The kernel value of CNTK and CNN-GP are calculated using the CuPy package. We write native CUDA codes to speed up the calculation of the kernel values. All experiments are performed on Amazon Web Services (AWS), using (possibly multiple) NVIDIA Tesla V100 GPUs. For efficiency considerations, all kernel values are computed with 32-bit precision.

One unique advantage of the dynamic programming algorithm for calculating CNTK and CNN-GP is that we do not need repeat experiments for, say, different values of $c$ in LAP and different depths. With our highly-optimized native CUDA codes, we spend roughly 1,000 GPU hours on calculating all kernel values for each dataset.

---

[4]We also tried kernel SVM but found it significantly degrading the performance, and thus do not include the results.

# E    TEST ACCURACY OF CNTK AND CNN-GP ON FASHION-MNIST

| c \ d | 5 | 8 | 11 | 14 |
|---|---|---|---|---|
| 0 | 92.25 (92.56) | 92.22 (92.51) | 92.11 (92.29) | 91.76 (92.17) |
| 4 | **93.76** (**94.07**) | 93.69 (93.86) | 93.55 (93.74) | 93.37 (93.58) |
| 8 | 93.72 (93.96) | 93.67 (93.78) | 93.50 (93.58) | 93.32 (93.51) |
| 12 | 93.59 (93.80) | 93.58 (93.70) | 93.35 (93.44) | 93.21 (93.40) |
| 16 | 93.50 (93.62) | 93.42 (93.63) | 93.27 (93.40) | 93.10 (93.25) |
| 20 | 93.10 (93.34) | 93.17 (93.49) | 93.20 (93.34) | 92.99 (93.18) |
| 24 | 92.77 (93.04) | 93.07 (93.44) | 93.11 (93.31) | 93.02 (93.21) |
| 28 | 92.80 (92.98) | 93.08 (93.42) | 93.12 (93.28) | 92.97 (93.19) |

Table 5: Test accuracy of CNTK on Fashion-MNIST.

| c \ d | 5 | 8 | 11 | 14 |
|---|---|---|---|---|
| 0 | 91.47 (91.81) | 91.96 (92.37) | 92.09 (92.60) | 92.22 (92.72) |
| 4 | 93.44 (93.60) | 93.59 (**93.79**) | **93.63** (93.76) | 93.59 (93.64) |
| 8 | 93.26 (93.16) | 93.41 (93.51) | 93.31 (93.52) | 93.39 (93.46) |
| 12 | 92.83 (92.94) | 93.07 (93.20) | 93.11 (93.15) | 92.94 (93.09) |
| 16 | 92.46 (92.51) | 92.58 (92.83) | 92.64 (92.92) | 92.68 (93.07) |
| 20 | 91.83 (91.72) | 92.35 (92.42) | 92.49 (92.79) | 92.51 (92.69) |
| 24 | 91.15 (91.40) | 92.10 (92.18) | 92.29 (92.60) | 92.41 (92.77) |
| 28 | 91.30 (91.37) | 92.03 (92.27) | 92.41 (92.79) | 92.41 (92.74) |

Table 6: Test accuracy of CNN-GP on Fashion-MNIST.

# F    VALIDATION ACCURACY OF CNTK AND CNN-GP ON CIFAR-10 AND FASHION-MNIST

| c \ d | 5 | 8 | 11 | 14 |
|---|---|---|---|---|
| 0 | 64.26 (68.42) | 64.47 (68.23) | 63.94 (67.80) | 63.29 (67.00) |
| 4 | 75.97 (78.87) | 75.89 (78.99) | 75.65 (78.56) | 75.40 (78.19) |
| 8 | 77.93 (80.65) | 77.90 (80.69) | 77.65 (80.41) | 76.92 (79.94) |
| 12 | 78.51 (80.73) | 78.47 (**80.85**) | 78.18 (80.57) | 77.71 (80.19) |
| 16 | 78.47 (80.39) | **78.69** (80.56) | 78.34 (80.17) | 77.74 (79.97) |
| 20 | 77.86 (79.69) | 77.81 (79.81) | 77.38 (79.55) | 76.88 (79.46) |
| 24 | 76.59 (78.12) | 76.80 (78.63) | 76.44 (78.79) | 76.18 (78.73) |
| 28 | 75.44 (77.08) | 76.15 (78.20) | 76.10 (78.30) | 75.95 (78.37) |
| 32 | 75.33 (76.99) | 76.04 (78.09) | 76.08 (78.27) | 75.99 (78.32) |

Table 7: Validation accuracy of CNTK on CIFAR-10.

| $c$ \ $d$ | 5 | 8 | 11 | 14 |
|---|---|---|---|---|
| 0 | 62.49 (66.63) | 64.25 (68.20) | 64.94 (69.01) | 65.35 (69.29) |
| 4 | 75.31 (78.59) | 76.05 (79.20) | 76.05 (79.17) | 76.20 (79.09) |
| 8 | 78.17 (81.02) | 78.53 (81.29) | 78.36 (81.20) | 78.05 (80.98) |
| 12 | 79.19 (81.38) | 79.08 (**81.66**) | 79.13 (81.52) | 78.90 (81.10) |
| 16 | **79.26** (81.18) | 79.24 (81.37) | 78.85 (81.33) | 78.82 (80.84) |
| 20 | 78.72 (80.61) | 78.72 (80.85) | 78.45 (80.60) | 78.08 (80.22) |
| 24 | 77.31 (79.01) | 77.59 (79.49) | 77.41 (79.56) | 77.26 (79.38) |
| 28 | 76.01 (77.60) | 76.60 (78.32) | 76.57 (78.76) | 76.86 (79.01) |
| 32 | 75.72 (77.54) | 76.42 (78.47) | 76.56 (78.94) | 76.63 (78.87) |

Table 8: Validation accuracy of CNN-GP on CIFAR-10.

| $c$ \ $d$ | 5 | 8 | 11 | 14 |
|---|---|---|---|---|
| 4 | 83.89 (85.76) | 83.13 (85.40) | 82.62 (84.95) | 82.02 (84.43) |
| 8 | 85.52 (87.59) | 84.88 (87.12) | 84.30 (86.69) | 83.84 (86.10) |
| 12 | **85.71** (**87.85**) | 85.32 (87.42) | 84.81 (87.02) | 84.26 (86.58) |
| 16 | 85.68 (87.76) | 85.19 (87.30) | 84.71 (86.83) | 84.47 (86.40) |
| 20 | 85.26 (87.11) | 84.91 (86.67) | 84.44 (86.40) | 84.09 (86.17) |

Table 9: Validation accuracy of additional feature extractor + CNTK on CIFAR-10.

| $c$ \ $d$ | 5 | 8 | 11 | 14 |
|---|---|---|---|---|
| 4 | 84.03 (86.16) | 84.21 (86.38) | 84.15 (86.33) | 83.98 (86.04) |
| 8 | 85.85 (87.94) | 85.87 (88.03) | 85.70 (87.87) | 85.49 (87.62) |
| 12 | 86.37 (**88.33**) | **86.38** (88.25) | 86.06 (88.12) | 85.69 (87.82) |
| 16 | 86.06 (88.27) | 86.21 (88.05) | 86.01 (87.87) | 85.58 (87.74) |
| 20 | 85.73 (87.71) | 85.79 (87.60) | 85.73 (87.54) | 85.27 (87.21) |

Table 10: Validation accuracy of additional feature extractor + CNN-GP on CIFAR-10.

| $c$ \ $d$ | 5 | 8 | 11 | 14 |
|---|---|---|---|---|
| 0 | 92.07 (92.30) | 92.08 (92.21) | 91.79 (91.99) | 91.51 (91.72) |
| 4 | **93.84** (**93.96**) | 93.82 (93.83) | 93.58 (93.74) | 93.40 (93.57) |
| 8 | 93.82 (**93.96**) | 93.80 (93.77) | 93.56 (93.71) | 93.37 (93.57) |
| 12 | 93.71 (93.83) | 93.60 (93.72) | 93.45 (93.58) | 93.41 (93.45) |
| 16 | 93.59 (93.73) | 93.39 (93.63) | 93.34 (93.53) | 93.21 (93.45) |
| 20 | 93.24 (93.44) | 93.29 (93.42) | 93.26 (93.30) | 93.19 (93.31) |
| 24 | 93.16 (93.28) | 93.21 (93.39) | 93.30 (93.32) | 93.22 (93.32) |
| 28 | 93.11 (93.23) | 93.21 (93.33) | 93.29 (93.29) | 93.28 (93.31) |

Table 11: Validation accuracy of CNTK on Fashion-MNIST.

| $c$ \ $d$ | 5 | 8 | 11 | 14 |
|---|---|---|---|---|
| 0 | 91.13 (91.43) | 91.57 (91.77) | 91.85 (91.92) | 91.94 (92.08) |
| 4 | 93.44 (93.55) | 93.57 (93.54) | **93.69** (93.68) | 93.58 (93.64) |
| 8 | 93.57 (93.67) | 93.51 (93.68) | 93.52 (**93.72**) | 93.44 (93.58) |
| 12 | 93.15 (93.36) | 93.49 (93.59) | 93.25 (93.52) | 93.23 (93.44) |
| 16 | 92.83 (92.84) | 93.01 (93.19) | 93.01 (93.27) | 92.95 (93.18) |
| 20 | 92.29 (92.45) | 92.60 (92.82) | 92.60 (92.93) | 92.78 (93.10) |
| 24 | 91.76 (92.04) | 92.28 (92.63) | 92.57 (92.86) | 92.58 (92.78) |
| 28 | 91.79 (92.00) | 92.32 (92.56) | 92.56 (92.77) | 92.70 (93.00) |

Table 12: Validation accuracy of CNN-GP on Fashion-MNIST.

## G  SETTING OF THE EXPERIMENT IN SECTION 6.3

The total number of training epochs is 80, and the learning rate is 0.1 initially, decayed by 10 at epoch 40 and 60 respectively. The momentum is 0.9 and the weight decay factor is 0.0005. In Figure 1, the blue line reports the average test accuracy of the last 10 epochs, while the red line reports the best test accuracy of the total 80 epochs. Each experiment is repeated for 3 times. We use circular padding for both convolutional layers and the BF layer. The last data point with largest $x$-coordinate reported in Figure 1 corresponds to GAP.

