# OpenReview forum: "Enhanced Convolutional Neural Tangent Kernels"
_ICLR.cc/2020/Conference — Reject_

### Official Review · AnonReviewer2 · 2019-10-14
**Official Blind Review #2**

**Rating:** 3

**Review:**

This paper shows that there is a one-to-one correspondence between pixel-shift based data augmentation and average pooling operations in CNN-NNGP/NTK based ridge regression. Interestingly, the authors show that standard average pooling + flatten can lead to a better performance than simple global average pooling. This paper further shows that using the data pre-processing step proposed in (Coates et al., 2011) can boost performance of CNN-NNGP/NTK based ridge regression by ~7% which allowed the authors to achieve classification accuracy in high 80s which is AFAIK SOTA on CIFAR-10 when not using learned representations.

My current assessment of the paper is “weak accept”. There are two main reasons why I am on the verge of recommending rejection of this paper: (1) I believe that the experiment evaluation is not done entirely correctly leading to inflation of the reported results (my guesstimate is by ~0.5-2%)---please see my “Major comments”. If this is not fixed, I am very likely to downgrade my score. (2) While the observation of the relationship between pixel-shifts and average pooling is very nice (which is why my current score is “weak accept”), it seems that most of the improvement comes from application of the pre-processing step of Coates et al. (2011) (seems like a ~7% improvement!). Given the large computational cost of CNN-NNGP/NTK (authors say about 1000 GPU hours), I wonder whether a simpler algorithm like some of the newer variants of boosting combined with the Coates et al. algorithm wouldn’t also perform at around 87-88% like CNN-NNGP/NTK (given the baseline 85-86% accuracy of the Coates et al. (2011) algorithm reported by the authors).


Major comments:

- Can you please clarify why you decided to give a new name (Box Blur) to standard average pooling? Why not just use the existing name?

- I believe that the way you report results in all the tables (i.e., tables 1-6) and the text based upon them is flawed. The right approach would be to select the hyperparameters “c” and “d” on a validation set, and then report the performance with these hyperparameters on the test set. While the experiments are somewhat rescued by the fact that you report results for (almost) all the possible hyperparameter settings (which allows us to see samples from the population distribution of the generalisation error), type-setting the best results in boldface and thus implying that these are valid estimates of the generalisation error is not appropriate since you are effectively selecting the best hyper-parameters on the test set! Unfortunately, I cannot accept these results to be published “as-is”. While re-running the experiments with hyperparameter selection on validation set is already a somewhat imperfect solution, I am not sure I can see a better way forward. However, I do understand that this could be prohibitively expensive in which case I would like to ask you to suggest an alternative solution please (of course, other reviewers are welcome to chime in as well)?!

- While most of the paper is about Local Average Pooling (LAP) and the equivalence between averaging and pixel shifts, the experimental results seem to show that most of the improvement comes from the use of Coates et al.’s preprocessing step. Could you please run the experiments in tables 3 and 4 with c=0 and c=32 to see what the effect of the preprocessing is without LAP?

- In sect.6.3, you say “Our experiment illustrates that even with a fixed last FC layer, using GAP could improve the performance of CNN, and challenges the conjecture that GAP reduces the number of parameters in the last fully-connected layer and thus avoids overfitting.” I am not sure I see why fixing the last FC layer should provide more convincing evidence than training it? I do not know the conjecture to which you refer but from your description, the overfitting without GAP should occur because the FC layer has more parameters than with GAP?! If this is true, then the overfitting would happen in the last layer (due to the large number of parameters) which you have (at least partially) prevented by not training it?! Can you clarify and also report the results of this experiment with all the layers trained please?

- In Appendix D, you say that you have used lambda = 10^{-5} for all configurations. How have you selected this particular value please? Do you have a sense of how far from optimal this value is for all the different configurations (or at least for NTK vs NNGP models---in my experience, the optimal setting between the two can differ quite a bit)?


Minor comments:

- In the abstract and throughout the paper, you claim that the cost of kernel regression is quadratic. AFAIK without any approximations, the cost is cubic (or O(n^{2.67}) to be more precise). Please clarify.

- In par.1 on p.1, you say “convolutional neural networks (CNNs) whose width (number of channels in convolutional layers) **goes to infinity**” (emphasis mine) and cite the Jacot et al. (2018) paper. AFAIK this paper only works with infinite networks but does not actually prove that **deep** networks of finite width (in each layer) converge to the NTK limit; IMHO you should cite the Allen-Zhu et al. (2018) and Du et al. (2018) papers from your references for that result. Based on p.2 (end of par.2 in sect.2), you seem to be aware of this distinction but cite Arora et al. (2019) instead of these two; I would suggest either citing Allen-Zhu et al. and Du et al. only, or citing all three as the Arora et al. paper came out later than the first versions of the other two paper which AFAIK already contained all the necessary derivations (even if the words “Neural Tangent Kernel” were not spelled out there).

- Also in par.1 on p.1, you say that Arora et al. (2019) was the first to provide an algorithm to compute the CNTK kernel which is a bit of a stretch given that both Garriga-Alonso et al. (2019) and Novak et al. (2019) have implemented the CNTK kernel in their experiments. AFAIK the claim in (Arora et al., 2019) is that they provided first **efficient** implementation of the CNTK-GAP kernel which should be made clearer in the next revision of your paper.

- On p.2, you say “These kernels correspond to neural networks where only the last layer is trained.” In reality, the correspondence is not exact for finite networks because the induced kernel will not be exactly equal to the one at the limit.

- Bottom of p.2, “Global Average Pooling (GAP) is proposed” -> “... was proposed”.

- Top of p.3, “..., and GAP is more robust” -> “..., and that GAP is more robust”.

- On p.3 in the “Padding Schemes” paragraph, do you mean to assume that the input image has only a single channel (not necessary later)?

- On p.4, I am slightly confused by your definition of the “augmented kernel”. Specifically, it does not seem K^G (x , x’) = K^G (x’, x) holds in general. Can you please clarify? If there’s no symmetry, I do not think it necessary to use a different name, but perhaps a clarifying note would be beneficial to the reader?!

- On p.5, fig.1 is too small when printed and one needs to use the computer screen to see what is depicted; given the amount of white space around, can you please try to make the images larger (you can perhaps also only include 2 or 4 images instead of 16 which will give you additional space)?

- On p.5, you say that for small “c”, circular padding will not create unrealistic images. Looking at fig.1b, it seems like the images are not as unrealistic as in fig.1a but human eye can still tell they are not realistic (potentially even more so with other images than the one selected for this figure). I am not sure whether there is a reason to assume this issue does not affect CNNs too?! Further, I am not convinced the motivation is correct in the first place given that the optimal “c” for CIFAR-10 is 12 which will presumably create clearly unrealistic images; perhaps it would be best to omit this motivation?!

- On p.6, you claim “Another advantage of LAP is that it **does not incur any extra computational cost**” (emphasis mine) while at the next line you say that there is a constant additional computational cost. Perhaps say that the extra computational cost is relatively small?

- It might be nice to swap tables 3 and 4 so at least the results for NNGP are next to each other. Even better would be the current table 3 was closer to table 1 to achieve the same effect for NTK.

- I am not sure I fully understand the description in sect.6.3: isn’t the number of channels on the input irrelevant after computation of the kernel in the first layer? In other words, why have you opted to use only 2,048 patches in your experiments and not 32,000 or 256,000 as used by Recht et al. (2019)? Do you have an estimate of how different could the performance of NNGP/NTK be with the larger number of features? Do you know what is the performance of Coates et al.’s algorithm with only 2,048 features? Relatedly, do you know how AlexNet would perform if its PCA data augmentation was replaced by the Coates et al.’s feature extractor?

**Experience Assessment:**

I have published in this field for several years.

**Review Assessment: Checking Correctness Of Derivations And Theory:**

I assessed the sensibility of the derivations and theory.

**Review Assessment: Checking Correctness Of Experiments:**

I assessed the sensibility of the experiments.

**Review Assessment: Thoroughness In Paper Reading:**

I read the paper at least twice and used my best judgement in assessing the paper.

---

> ### Author Response · Authors · 2019-11-15
> **Response**
>
> Thank you for your review. We have revised our paper according to your comments. Please find our response to your comments below.
> 1.	We acknowledge this is not a new operation. We now use the name box filtering according to [1].
> 2.	We have added cross-validation in our experiments. See the first paragraph in Section 6.1 for the detailed procedure of cross-validation. We have underlined the accuracy on test data that corresponds to the best hyper-parameter via cross-validation. We remark that we still achieve 88.91% accuracy on CIFAR-10 with cross-validation.
> 3.	We are sorry that we do not have computational resources to run $c=0$ and $c=32$ during the rebuttal period since it requires roughly 1,000 GPU hours. Note that in the setting where features are predefined without using the data, then LAP gives 2-3% improvement over GAP.
> 4.	Sorry for the confusion. The conjecture refers to the sentence in the original paper that proposed GAP (Lin et al., 2013), which stated that one possible reason that GAP improves the performance is that CNN with GAP has fewer training parameters than CNN without GAP. Our point is that since we fixed the last fully-connected layer, CNNs with GAP or without GAP have the same number of training parameters but GAP still improves the performance, so the benefit of GAP may not be explained by the fact that it leads to fewer training parameters.
> 5.	We choose the smallest $\lambda$ so that solving kernel regression is numerically stable across all settings.
>
> [1] Richard Szeliski. Computer vision: algorithms and applications. Springer Science & Business Media, 2010.
>
>
>
> For minor comments:
> 1.	For CIFAR-10, the bottleneck of using CNN-GP and CNTK is not solving kernel regression but computing the kernel values. The time complexity is $O(p^2 n^2)$ where p is the number of pixels in each image and $n$ is number of data point. We have added a clarification in the paper.
> 2.	Both Garriga-Alonso et al. (2019) and Novak et al. (2019) have implemented the CNN-GP kernel rather than the CNTK kernel. We have changed the first paragraph to make it clear.
> 3.	We have slightly changed the definition of the “augmented kernel”, so that $K^{\mathcal{G}}$ is a kernel even when $K$ is not invariant under $\mathcal{G}$. If $K$ is invariant, the definition remains the same.
> 4.	For Figure 1, we have enlarged its size and moved it to appendix.
> 5.	When “c” is as large as 12, it does create unrealistic images, but certainly with much smaller probability than full translation. In fact, whether the image is unrealistic is not exactly what we care about. As long as an augmented image much closer to its real class than other classes (e.g., Figure 1.b is much more like a “truck” than an “automobile”/”airplane”/”bird”), it will likely improve the robustness of the classifier, and thus improve the accuracy. Therefore, we should choose a proper value for $c$ to get as many “helpful” augmented samples as possible.
> 6.	If the input images has $c$ channels and $p$ pixels, calculating the kernel value in the first layer requires $O(cp^2)$ time (see the formula for $K^{(0)}$ in Appendix A). Thus, with $n$ input images, the total cost would be $O(n^2cp^2)$. With very large value of $c$ this would be even more expensive than calculating the rest part of the kernel values. In our experience, with $c = 2048$ channels, calculating the kernel value in the first layer requires roughly 750 GPU hours on NVIDIA Tesla V100. If we take $c = 256, 000$ for example, this will require roughly 100, 000 GPU hours. We are sorry but this is well-beyond our computational resources available. We are planning to perform the other experiments requested by the reviewer after getting more computational resources.

---

### Official Review · AnonReviewer3 · 2019-10-23
**Official Blind Review #3**

**Rating:** 6

**Review:**

This paper builds on recent developments of CNN-GP and CNTKs in multiple fronts obtaining significant performance boost on CIFAR-10 dataset (and some mild boost on Fashion-MNIST). One way is by usage of Local Average Pooling (LAP) layers which interpolates between Global Average Pooling (GAP) and no Pooling layer. The authors also introduce flip data augmentation by doubling the dataset. With the help of additional feature extractor, this paper obtained 89% classification accuracy on CIFAR-10 which is the best among methods not using trained neural networks.

The discussion on section 4 regarding augmented kernel and data augmentation is quite clear and revealing. It’s unfortunate that the flip augmentation could not be introduced in kernel level. It would be interesting for future work to find kernel operation similar to GAP that encodes symmetries of the dataset.

While the paper is clearly written and the results are strong, there are few criticisms I’d like to address and hope the authors address.

AFAIK both GAP and LAP for CNN-GP are already introduced and analyzed in [1]. It seems best results on CIFAR-10 all comes from CNN-GP (with without flip augmentation, with and without using extra feature extractor), and I think the authors should properly credit [1] for GAP/LAP in convolutional kernels. It’s fair that this paper along with [2] was able to efficiently implement and scale up to  full CIFAR-10 dataset and demonstrated pooling layer’s full potential for kernels corresponding to infinitely wide CNNs. Also in this regard the title could be misleading. It’s strange to have paper’s strongest result is based on CNN-GP while the title only mentions CNTK.

As the author’s mention in the paper, Box Blur is just an average pooling operation. This is already widely use by practitioners(e.g. [3]) and I don’t understand how author’s claim: “This operation also suggests a new pooling layer for CNNs which we call BBlur”

Few question/comments:

Best parameters for trained CNN’s BBlur c is smaller than best c values for kernels, do authors understand the cause of discrepancy?

It would benefit the research community if authors could share code to generate the CNN-GP Kernels / CNTKs with LAP.  Also I would encourage authors to share actual numerical values of kernel matrix for other research groups to analyze and encourage reproducibility.


[1] Novak et al., Bayesian Deep Convolutional Networks with Many Channels are Gaussian Processes, ICLR 2019
[2] Arora et al., On Exact Computation with an Infinitely Wide Neural Net, NeurIPS 2019
[3] Huang et al., Densely Connected Convolutional Networks, CVPR 2017


**Experience Assessment:**

I have published in this field for several years.

**Review Assessment: Checking Correctness Of Derivations And Theory:**

I carefully checked the derivations and theory.

**Review Assessment: Checking Correctness Of Experiments:**

I carefully checked the experiments.

**Review Assessment: Thoroughness In Paper Reading:**

I read the paper thoroughly.

---

> ### Author Response · Authors · 2019-11-15
> **Response**
>
> Thank you for your review. We have revised our paper according to your comments. Please find our response to your comments below.
> 1.	We do agree GAP is also introduced in [1] and we do not claim we invented GAP. In fact, GAP is first proposed in Lin et al. (2013) and is a standard component in modern CNNs. For LAP, we do not think it is introduced in [1]. At least [1] does not define such an operation explicitly. Moreover, although [1] shows that CNN-GP with GAP is invariant to translations, in this paper we establish a formal connection between GAP and full translation data augmentation, which is novel and does not appear in [1].
> 2.	We have changed our title to reflect the fact that our paper deals with both CNN-GP and CNTK.
> 3.	We acknowledge this is not a new operation. We now use the name box filtering according to [4].
> 4.	For CNN we used many standard tricks including batch norm, weight decay and momentum. Note CNTK corresponds to CNN without using these tricks and are trained via gradient flow.
> 5.	We have provided a link to our current implementation. We will further clean up the codes and make it public after acceptance. We will share the kernel values as well.
>
> [4] Richard Szeliski. Computer vision: algorithms and applications. Springer Science & Business Media, 2010.

---

### Official Review · AnonReviewer1 · 2019-10-23
**Official Blind Review #1**

**Rating:** 6

**Review:**

This paper considers architectures that do not involve learning (up to the classification layer) and tries to improve their accuracies. They're based on CNTK and CNN-GP works. This is purely a numerical paper and its contribution is to show that despite being not learned, the obtained representations are competitive with supervised neural networks.

Overall, despite the fact if I find this numerical result interesting, I found too many flaws to justify its acceptance. (fine tuning on the test set, lack of comparison with the state of the art...)

Pros:
- Good numerical performances.

Cons:
- Given the claim in the abstract about accuracies, it should be pointed out that:
* in the unsupervised setting, with a kernel engineering method, you can obtain ~86% on cifar10 (cf https://arxiv.org/abs/1605.06265 )
* in the no-data(up to a linear model) setting, it is possible to get ~82% on cifar10 with the scattering networks (cf https://arxiv.org/abs/1412.8659 )
Those two works are also mainly empirical, and thus some accuracies of this paper should be compared to them.
- There is a significant amount of experiments (table 1/2/3/4). While this should have been a positive aspect of the paper, I noticed that the accuracies reported here are computed from the test set. A validation set should have been used with a careful cross-validation. I'm aware this is a standard practice in deep learning, yet here it seems obvious to me that some hyper parameters have been fine-tuned on the testing set.
- Section 4: isn't it a rephrasing of (Dao et al, 2018)? (which is cited) I think this should be clearly stated.
- Section 5: The paper cites the Local Average Pooling as a "new operation", but this is clearly standard in the literature. "Boxblurring" has always been named average pooling in deep learning, low-pass filtering in signal processing. It was used before researchers employ a stride of 2 in convolutions. A similar pooling is also present in https://arxiv.org/abs/1605.06265
- I'm nicely surprised that the authors didn't encounter any significant conditioning issues. Would it be possible to show the spectrum of the kernel? This could be commented.
- Nothing about the future release of the code is indicated.

Minor:
- I find the Figure 1 is not informative to the reader.

Post-discussion:
The revision clarifies all my concerns and this work is likely to induce interesting discussions.

**Experience Assessment:**

I have published one or two papers in this area.

**Review Assessment: Checking Correctness Of Derivations And Theory:**

N/A

**Review Assessment: Checking Correctness Of Experiments:**

I carefully checked the experiments.

**Review Assessment: Thoroughness In Paper Reading:**

I read the paper thoroughly.

---

> ### Author Response · Authors · 2019-11-15
> **Response**
>
> Thank you for your review. We have revised our paper according to your comments. Please find our response to your comments below.
> 1.	In the related work section, we have added discussion on SOTA results in the unsupervised learning and in the no-data setting.
> 2.	We have added cross-validation in our experiments. See the first paragraph in Section 6.1 for the detailed procedure of cross-validation. We have underlined the accuracy on test data that corresponds to the best hyper-parameter via cross-validation. We remark that we still achieve 88.91% accuracy on CIFAR-10 with cross-validation.
> 3.	Our results in Section 4 is not a restatement of or implied by (Dao et al. 2018). In our paper we consider two types of data augmentation. The first type is enlarging the training dataset (as used in practice), and the second type is averaging the prediction on new training samples obtained by applying different transformation on the original training samples. These two methods are in general very different, except for unrealistic cases when the loss function is almost linear, as discussed in (Dao et al., 2018). (Dao et al., 2018) also assumes the augmented images from the same image has very small variance such that the quadratic term of Taylor expansion of the loss vanishes, which is not the case for operations studied in this paper. For instance, horizontal flip could induce large variance. Even if the objective values given by these two type augmentation methods might be close, the gradients can still be very different, and thus could result in different trajectories and solutions. In this paper, we give a much stronger result in the case of kernel regression (note the loss is l2 and thus non-linear). We prove that when the transformations used to generate augmented samples form a group, solving kernel regression using the above two types of data augmentation gives *exactly* the same solution, which explains the success of Global Average Pooling.
> 4.	We acknowledge this is not a new operation. We now use the name box filtering according to [1].
> 5.	Computing the spectrum requires performing SVD on a 50000 by 50000 matrix, which is computationally infeasible given our computational resources.
> 6.	We have provided a link to our current implementation. We will further clean up the codes and make it public after acceptance.
> 7.	We have moved Figure 1 to appendix.
>
> [1] Richard Szeliski. Computer vision: algorithms and applications. Springer Science & Business Media, 2010.

---

> > ### Comment · AnonReviewer1 · 2019-11-15
> > **Thanks for the rebuttal**
> >
> > Dear authors,
> >
> > 1. Thanks for the clarification.
> > 2. Thanks for the clarification.
> > 3. So, as far as I understood, (Dao et al) consider data augmentation which are linearized. As all the layers of your architectures are (linearly) covariant with the action of translation, it implies that the action of translations on the sample $x$ is a linear action on the obtained representation. This is exactly the setting of the Section 4.1 of (Dao et al), if a subset of translations is uniformly sampled. In other words, if a group action is linear, averaging along an orbit of the group leads to a linear operator. I agree that this setting is different for the flips, yet this is a group with simply 2 elements...  Am I incorrect?
> > 4. Thanks for the clarification.
> > 5. I partially agree: I think it is still computationally tractable (maybe not on standard academic resources), however approximate methods exist. I think this would have been interesting, as you observed that directly solving the regression (which incorporates a regularization) allows to obtain good performances: it is surprising given that there is no supervision. Thanks however for the clarification.
> > 6. Thanks.
> > 7. OK.
> >
> > I will revise my review. Thank you very much for your rebuttal.

---

> > > ### Author Response · Authors · 2019-11-15
> > > **Thank you for your reply**
> > >
> > > Thank you for your reply!
> > >
> > > On difference from (Dao et al. 18):
> > > To clarify, (Dao et al. 18) assumes the loss is linear (second order term vanishes), and has no restriction on the transformations. We do not need to assume loss is linear (we considered l2 loss), but we require transformations to form a group.
> > >
> > > Therefore, the results in (Dao et al. 18) cannot be applied to our setting.
> > >
> > >
> > > On linear actions: Both the result in (Dao et al. 18) and our result do not require the linearity of actions, although both translation and flip are indeed linear (viewed as functions on the image space).

---

### Author Response · Authors · 2019-11-15
**General Response and Revision Summary:**

We thank all reviewers for their constructive comments! All major changes in our paper are marked in red. We made the following main changes in our revision.
1.	We changed the title according to suggestion from Review #3.
2.	We added more detailed discussions on previous work on methods that are not trained networks in the last paragraph of Section 2.
3.	We have added validation accuracy in Appendix F. The cross-validation details are in Section 6.1.
4.	We now use the name box filtering for the operation on CNN according to [1].
5.	We have provided a link to our current implementation


[1] Richard Szeliski. Computer vision: algorithms and applications. Springer Science & Business Media, 2010.

---

### Decision · Program_Chairs · 2019-12-19

**Decision:**

Reject

**Comment:**

This paper was assessed by three reviewers who scored it as 6/3/6.
The reviewers liked some aspects of this paper e.g., a good performance, but they also criticized some aspects of work such as inventing new names for existing pooling operators, observation that large parts of improvements come from the pre-processing step rather than the proposed method, suspected overfitting.  Taking into account all positives and negatives, AC feels that while the proposed idea has some positives, it also falls short of the quality required by ICLR2020, thus it cannot be accepted at this time. AC strongly encourages authors to go through all comments (especially these negative ones), address them and resubmit an improved version to another venue.